# Repellent Effects of Coconut Fatty Acid Methyl Esters and Their Blends with Bioactive Volatiles on Winged *Myzus persicae* (Sulzer) Aphids (Hemiptera: Aphididae)

**DOI:** 10.3390/insects15090731

**Published:** 2024-09-23

**Authors:** Félix Martín, Pedro Guirao, María Jesús Pascual-Villalobos

**Affiliations:** 1Instituto Murciano de Investigación y Desarrollo Agrario y Medioambiental (IMIDA), c/Mayor s/n, 30150 La Alberca, Murcia, Spain; mjesus.pascual@carm.es; 2Escuela Politécnica Superior de Orihuela (EPSO), Universidad Miguel Hernández, Carretera de Beniel km. 3.2, 03312 Orihuela, Alicante, Spain; pedro.guirao@umh.es

**Keywords:** nanoemulsions, farnesol, *(E)*-anethole, pepper plants, choice bioassay, virus-transmitting sucking pests

## Abstract

**Simple Summary:**

Aphids (Hemiptera: Aphididae) are among the most damaging pests for crops, due to the direct damage they cause and their ability to transmit viral diseases. Direct control of viral diseases is not possible, and hence, it is necessary to avoid the arrival of vectors to the crop. The objective of this study is to test whether spraying nanoemulsions of botanical products repel winged individuals of *Myzus persicae* (Sulzer) in a bioassay in culture chambers. Some of the bioactive volatiles tested, such as farnesol (similar to the aphid alarm pheromone), *(E)*-anethole (the major compound in anise and fennel essential oils) and a compound derived from coconut oil, have shown repellent effects both individually and in blends. This is valuable information for further research.

**Abstract:**

*Myzus persicae* (Sulzer) (Hemiptera: Aphididae) is one of the most important aphid crop pests, due to its direct damage and its ability to transmit viral diseases in crops. The objective is to test whether spraying nanoemulsions of botanical products repels winged individuals of *M. persicae* in a bioassay in culture chambers. The bioactive volatiles were applied on pepper plants at a dose of 0.2% alone or at 0.1% of each component in blends. A treated plant and a control plant were placed at each side of an entomological cage inside a growth chamber. The winged individuals were released between the plants, in a black-painted Petri dish suspended by wires in the upper half of the cage. The most repellent products were farnesol (repellency index, RI = 40.24%), *(E)*-anethole (RI = 30.85%) and coconut fatty acid methyl ester (coconut FAME) (RI = 28.93%), alone or in the following blends: farnesol + *(E)*-anethole + distilled lemon oil (RI = 36.55%) or *(E)*-anethole + distilled lemon oil + coconut FAME (RI = 30.63%). The observed effect of coconut FAME on aphids is the first report of this product having a repellent effect on a crop pest. Repellent substances for viral disease vectors should be further investigated to develop new strategies for plant protection.

## 1. Introduction

According to the final report issued by the Food and Agriculture Organization of the United Nations in 2021 [1] following the commemoration of the International Year of Plant Health 2020, agricultural diseases and pests are responsible for as much as 40% of losses in global food crop production, and for losses in agricultural commodity trade worth more than USD 220 billion each year.

One of the most important crop pests are aphids (Hemiptera: Aphididae). There are more than 5000 species of Aphididae in the world [2]. Of these, about 450 species have been recorded from crop plants [3]. Direct damage is due to aphid feeding and the honeydew they secrete, which is colonised by saprophytic fungi that reduce plant photosynthetic capacity and growth, and reduce the commercial value of crops, among other effects. On the other hand, they may cause major indirect damage in that they are the main vectors of plant viruses (circulative and non-circulative) [4,5]. Aphid-borne viruses belong to 19 of the 70 recognized genera of plant pathogenic viruses and comprise approximately 275 species of viruses that affect plants [6]. The green peach aphid *Myzus persicae* (Sulzer) is among the 14 aphid species of most agricultural importance and is highly efficient as a virus vector. It is the principal pest of protected pepper crops in southeastern Spain [7,8,9]. Specifically in pepper, *M. persicae* very effectively transmits CMV (cucumber mosaic virus, *Cucumovirus*) [10] and PVY (potato virus Y, *Potyvirus*) [11] in a non-persistent way, and PeVYV (pepper vein yellows virus, *Polerovirus*) [12] in a persistent way, among other viruses.

Direct control of viral diseases is not possible, and hence, it is necessary to anticipate the problem by implementing preventive measures such as the use of healthy or resistant plant material, weed control, the elimination of diseased plants and strict compliance with hygiene measures, as well as the monitoring of greenhouse enclosures. All of these factors can play an important role in the spread of viruses and allow for some degree of control, but for more effective prevention of viral diseases transmitted by insects, it is essential to avoid the arrival of the vector to the crop since the acquisition and transmission times of certain types of viruses are very short [13]. Curative treatments once aphids have settled on a crop are ineffective in preventing virus transmission, and therefore the use of phytosanitary products, with a repellent mode of action, is especially important for preventing the arrival of winged aphids to crops.

Winged aphids migrate looking for host plants to colonise. In recent years, winged aphids have been migrating earlier and earlier due to climate change and unusual periods of mild temperatures during the winter [14]. Winged aphid colonies are now seen as early as in the initial stages of cultivation in the organic pepper crops of Campo de Cartagena (Murcia, southeastern Spain), and this is a real problem.

Aphids locate plant material mainly by colour, but also by odour. Probably due to the evolutionary process, winged individuals have more developed olfactory antennal receptors than wingless individuals [15]. In these olfactory receptors, called rhinariums, odorant-binding proteins (OBPs) are concentrated, which participate in the perception of odours in the antennae. Electroantennography (EAG) has been shown to detect a response to a wide variety of volatile organic compounds (VOCs) in plants. Nonetheless, the role of these olfactory organs in aphid–plant interaction is still poorly understood [16].

Several studies have been carried out to identify VOCs such as ethers, phenols, monoterpenes and sesquiterpenes that repel aphids of horticultural importance. Most of these volatile compounds are found in the essential oils of plants which are extracted from various plant organs (roots, wood, leaves, flowers, fruits and seeds). They have historically been used in the food and perfumery industry, but not so much in agriculture [17,18,19]. Some studies show that essential oils and their compounds affect the physiological, nutritional and behavioural processes of insects. In particular, they cause mortality, are antifeedants, growth regulators and oviposition deterrents, reduce fertility, have ovicidal and fungicidal properties and/or prevent root penetration by nematodes [18,20,21,22,23,24].

It has also been shown that two species of *Ocimum* sp., associated with plants of *Amaranthus hybridus* L., had repellent effects on apterous individuals of *Aphis craccivora* Koch, *A. fabae* and *M. persicae* [25]. Furthermore, in olfactometry tests with apterous individuals of *M. persicae*, the repellent effects of basil plants (*Ocimum basilicum* L.) and also of *Tagetes patula* L. have also been demonstrated, when the volatiles were released individually. It has been found that eugenol and *(E)*-β-farnesene had good repellent properties against *M. persicae* [26]. Notably, *(E)*-β-farnesene was found to be the main component of the alarm pheromone of many aphids [16].

It has been reported that oils of rosemary, thyme, lavender, peppermint, ginger, white pepper, black pepper, carrot seeds, cardamom, bitter orange, citronella, celery seeds, cedarwood and laurel, as well as some compounds present in rosemary oil such as linalool, D,L-camphor and α-terpineol, showed repellent activity in an olfactometer test with winged individuals of *M. persicae* [27,28]. In addition, rosemary and ginger oils repelled winged forms of *M. persicae* and inhibited the transmission of potato virus Y (PVY, *Potyviridae*) in a tobacco plants greenhouse assay. Other potentially repellent substances for *M. persicae* were tested [29], and findings showed that crude fish oil, a summer mineral oil, and refined soy and rapeseed oils prevented the settlement of wingless individuals of *M. persicae* on pepper leaf discs, with fish oil having the greatest repellent effect. Nonetheless, in a trial with winged individuals (released into entomological cages), they did not observe significant differences in aphid settlement between treated and control pepper plants.

Concerning the effects on winged individuals of other aphid species [28], it has been demonstrated that rosemary oil repelled *Aphis gossypii* Glover and *Macrosiphum euphorbiae* (Thomas), and ginger oil showed significant repellency against *A. craccivora* Koch, *M. euphorbiae* and *Capitophorus formosartemisiae* (Takahashi). Furthermore, a test with winged individuals of *Rhopalosiphum maidis* (Fitch) released in a box containing untreated control tiles and others treated with β-citronellol, oils from several species of *Artemisia* sp., farnesol, geraniol, linalool, or *Achillea millefolium* L. oil (known as yarrow or milfoil) has been carried out, and it has reported promising repellent results [30]. In addition, it has been demonstrated that the essential oils of anise, basil and *Cymbopogon* sp., and several pure compounds such as *(E)*-anethole, geraniol, farnesol and *(Z)*-jasmone, inhibit the settlement of wingless individuals of *M. persicae* and *M. euphorbiae* in Petri dish choice bioassays with pepper leaves, with farnesol and *(Z)*-jasmone being the most repellent for both wingless and winged forms of both aphid species in olfactometry assays [31]. Recently, it has been reported that the essential oils of three Lamiaceae plants (*Mentha arvensis* L., *Mentha x piperita* L. and *Lavandula angustifolia* Miller) significantly repelled winged individuals of *A. gossypii* in a Y-tube olfactometer experiment [32].

After conducting a literature search, we found studies on substances that are repellents against other types of insects. In particular, compounds derived from coconut oil have been shown to have repellent properties against a broad array of blood-sucking arthropods including flies, ticks, bed bugs and mosquitoes [33].

Essential oils and their compounds, being oily substances, are frequently insoluble in water, and they exhibit rapid environmental degradation, high volatility and thermal decomposition and/or evaporation due to their poor physicochemical stability, which significantly decreases their activity [20,34,35,36]. Other types of substances such as surfactants, emulsifiers, or adjuvants, which provide greater protection of the active ingredient against rapid degradation or volatilization, may be able to improve the bioavailability of the active compounds for longer periods of time [37]. The stability of the mixture depends on the particle size in addition to its composition: if particles are within the nanometric range (<1 μm), the impact of Brownian movement is reduced and therefore, aggregation is less frequent [38]. Oil-in-water nanoemulsions are 200 to 1000 nm diameter oil droplets dispersed in aqueous media (surrounded by the surfactant) [39]. In recent years, O/W emulsions have attracted much attention due to the need to reduce toxic and flammable solvents from conventional phytosanitary formulations and replace them with substances of natural or plant origin. Plant-based substances have been used since ancient times as insecticides or adjuvants to fat-soluble molecules [40].

The mode of action of plant-based products is not well known. Studies focus on mortality, control of insecticide resistance and behavioural responses related to interference with virus transmission, but rarely on repellency. Therefore, the objective of this study was to test whether spraying nanoemulsions of bioactive volatiles (alone and as mixtures) on pepper plants repelled winged individuals of *M. persicae*, thereby preventing the arrival of this vector to the crop.

## 2. Materials and Methods

### 2.1. Insects and Plants

To evaluate the repellent and reproductive effects of the bioactive volatiles on winged aphids, pepper plants were grown in growth chambers at the Instituto Murciano de Investigación y Desarrollo Agrario y Medioambiental (IMIDA, Murcia, Spain) under controlled conditions at a constant temperature of 23 °C and with a light/dark cycle of 16:8 h.

The aphids used in these assays were adult winged individuals of the species *M. persicae* from cultures at IMIDA (Murcia, Spain), reared under controlled conditions at a constant temperature of 23 °C and with a light/dark cycle of 16:8 h. The *M. persicae* population was originally collected from a pepper crop in Campo de Cartagena (Murcia, Spain) and had been maintained on pepper at IMIDA since 2016.

The nymphs found in this experiment were born during the choice bioassay after the released winged aphids settled on the pepper plants.

### 2.2. Bioactive Volatile Products

The bioactive volatiles used were *(E)*-anethole, farnesol (Sigma-Aldrich^®^, San Louis, MI, USA), distilled lemon oil (Citromil S.L., Santomera, Murcia, Spain) and waxy starch-coconut fatty acid methyl ester composite (code 19391-122), hereinafter coconut FAME, and this product was provided by the researchers James A. Kenar and Steven C. Cermak (National Center for Agricultural Utilization Research, Agricultural Research Service, United States Department of Agriculture, Peoria, IL, USA). Another bioactive used was the product mixture formulated by Idai Nature S.L. (La Pobla de Vallbona, Valencia, Spain) using an undisclosed method.

The distilled lemon oil is composed of 71.1% limonene, 11.5% β-pinene, 8.2% γ-terpinene, 2.1% α-pinene + α-thuyene, 0.7% geraniol, 0.5% geraniol acetate and 0.4% neral [41]. The coconut FAME formulation is composed of 88.42% water, 6.98% waxy starch/pectin and 4.60% coconut fatty acid, and the Idai Nature S.L. product mixture contains equal amounts of *(Z)*-jasmone, citral, distilled lemon oil, farnesol and *(E)*-anethole, at a dose of 0.2% of the total of the product.

### 2.3. Preparation of Oil-in-Water (O/W) Nanoemulsions

Oil-in-water (O/W) nanoemulsions were prepared using a high-speed dispersion machine IKA^®^ Labor-Pilot Controller 2000/4 (IKA-Werke GmbH & CO. KG, Staufen, Germany) programmed to work each batch of 100 mL for 10 min at a rotor speed of 7941 rpm and a cooling temperature of 15 °C, which prevents overheating of the machine, maintaining the temperature in the mixing chamber at 20 ± 5 °C and thus preventing losses of the bioactives by volatilisation during the process.

For the experiment, the O/W nanoemulsions were formulated at 0.2% alone or at 0.1% in blends of more than one product, with Tween^®^ 80 (1:2, and for the blends, 1:1:2, 1:1:1:2, 1:1:1:1:2,) in distilled water, and used immediately for spraying.

### 2.4. Repellent and Reproductive Effects of Bioactive Volatile Nanoemulsions on Winged Individuals of M. persicae on Pepper Plants

Bioactive volatile nanoemulsions of *(E)*-anethole, farnesol, distilled lemon oil and coconut FAME, at 0.2% or 0.1% in blends of more than one product, with Tween^®^ 80 in distilled water or the product mixture diluted to 0.2%, were sprayed onto pepper plants 25–30 cm high and at a BBCH growth stage of 501 (first flower bud visible) with a handheld sprayer (Polita 7, Matabi; Grupo Goizper, Gipuzkoa, Spain) at a rate of approximately 100 mL/plant. Plants were then placed into a 100 × 55 × 70 cm entomological cage inside a growth chamber. A treated plant was placed at one end of the cage and a control plant (treated with water) at the other, the plants being separated by a distance of 70 cm. The choice test was conducted under controlled conditions at a constant temperature of 23 °C and a with light/dark cycle of 16:8 h. In each cage, 25 winged aphids were released from a Petri dish placed in the middle of the cage, suspended about 35 cm high with the help of wires attached to the wooden frame of the cage (Figure 1). The Petri dish was previously painted black to avoid providing visual cues for aphids when choosing a plant, enhancing the odour cue. After 24 h, the winged individuals and nymphs on each plant were recorded. The aphids and plants were renewed every day. The positions of the control plant and the treated plant were alternated in each cage and in each replicate to avoid bias in the choices.

In each replicate, six cages were used. A total of four replicates (applications) were performed per treatment per week (Monday to Thursday). Each treatment was evaluated in the same week. For each treatment, a total of 24 repetitions (cages) were carried out, using 24 control plants and 24 treated plants, and a total of 600 winged aphids were released (25 per cage).

The repellency index (%) (RI) was calculated after 24 h using the following formula [42]:(1)$$\mathrm{RI}\;(\%)=\left(1-\frac{T}{C}\right)\times 100,$$
where T is the number of aphids on the treated surface and C is the number of aphids on the control.

### 2.5. Statistical Analysis

For each treatment, the distribution of choices (expressed as a percentage) was calculated, i.e., the number of aphids that had chosen the treated plant or the number of aphids that had chosen the control plant divided by the total number of aphids that had chosen any plant. For statistical analysis, R software R version 4.2.2 was used [43]. The statistical significance of differences between each treatment and its control were assessed using an exact two-sided binomial test (* *p* < 0.05; ** *p* < 0.01; *** *p* < 0.001; ns *p* > 0.05) with Clopper–Pearson 95% confidence intervals.

The average number of nymphs per female 24 h after the release of winged individuals of *M. persicae* was also counted for each treatment and its control. The *t*-test for paired samples was used for mean comparison, with a *p*-value ≤ 0.05 set as the threshold for statistical significance.

## 3. Results

### 3.1. Choice Distribution of Winged Individuals of M. persicae on Pepper Plants

Table 1 and Figure 2 show the choice distribution (treated (T) or control (C) plant) 24 h after the release of winged individuals of *M. persicae* in cages with pepper plants. The overall average percentage of individuals that made a choice was 78.1%, indicating that the methodology used to conduct this assay was appropriate. The percentage of choice in all treatments was always greater than 40% and therefore, none of the treatments were excluded from the analysis.

Most treatments caused repellency (Table 1 and Figure 2). The most active compounds, with a very highly significant probability (*p* < 0.001) according to an exact two-sided binomial test, were farnesol (T = 147 individuals; 37% vs. C = 246 individuals; 63%), *(E)*-anethole (T = 195 individuals; 41% vs. C = 282 individuals; 59%) and coconut FAME (T = 172 individuals; 42% vs. C = 242 individuals; 58%). The mixtures that provided the best results were farnesol + *(E)*-anethole + distilled lemon oil (T = 184 individuals; 39% vs. C = 290 individuals; 61%) and *(E)*-anethole + distilled lemon oil + coconut FAME (T = 154 individuals; 41% vs. C = 222 individuals; 59%), as well as the five-product mixture treatment (T = 175 individuals; 38% vs. C = 288 individuals; 62%).

The binary mixtures of farnesol + *(E)*-anethole (T = 156 individuals; 42% vs. C = 219 individuals; 58%) and farnesol + distilled lemon oil (T = 177 individuals; 42% vs. C = 245 individuals; 58%), the ternary mixture of farnesol + distilled lemon oil + coconut FAME (T = 191 individuals; 44% vs. C = 247 individuals; 56%) and the quaternary mixture of farnesol + *(E)*-anethole + distilled lemon oil + coconut FAME (T = 200 individuals; 43% vs. C = 264 individuals; 57%) also showed highly significant levels of repellency (*p* < 0.01). Binary mixtures without farnesol seemed to be less repellent (Figure 2). In the treatments with just distilled lemon oil or the binary mixtures of farnesol + coconut FAME and distilled lemon oil + coconut FAME, differences between the number of aphids found on control and treated plants were non-significant, indicating that they had no repellent or attractant effects (Figure 2).

The repellency index (RI) is shown in Table 1, expressed as a percentage (%). The treatment with the highest RI was farnesol (RI = 40.24%), followed by the mixture of product treatment (RI = 39.24%) and the mixture of farnesol + *(E)*-anethole + distilled lemon oil (RI = 36.55%), while *(E)*-anethole and the mixture of *(E)*-anethole + distilled lemon oil + coconut FAME both provided RIs of around 31%. Finally, another treatment with interesting repellent effects but with a slightly lower RI was coconut FAME (RI = 28.93%).

In short, the most repellent products were found to be farnesol, coconut FAME and *(E)*-anethole if applied alone, and the mixtures of farnesol with *(E)*-anethole alone or also with distilled lemon oil, and *(E)*-anethole with coconut FAME and distilled lemon oil. Distilled lemon oil alone showed no activity, but it enhanced the repellent properties of the blends. With the mixture of products (farnesol, citral, *(E)*-anethole, *(Z)*-jasmone and distilled lemon oil), the results were also very good.

### 3.2. Reproduction of Winged Individuals of M. persicae on Pepper Plants

Table 2 shows the average number of nymphs per female 24 h after the release of winged individuals of *M. persicae* in cages with pepper plants (control and treated). Significant differences were only observed for the mixture with farnesol + distilled lemon oil + coconut FAME (*p* = 0.031), with the average number of nymphs on the treated plants (0.65 nymphs/female per day) being lower than that found in the control plants (0.97 nymphs/female per day). The treatments with the lowest mean were distilled lemon oil (0.40 nymphs/female) followed by *(E)*-anethole + distilled lemon oil + coconut FAME (0.44 nymphs/female), but differences with the corresponding controls did not reach significance in any cases. On the contrary, *(E)*-anethole alone stood out for being the treatment with which aphids reproduced in the largest numbers (1.16 nymphs/female per day), followed by the mixture of distilled lemon oil + coconut FAME (0.99 nymphs/female).

## 4. Discussion

Since the repellency of winged aphids is not well studied in the literature, nor is the repellent effect of nanoemulsions of bioactive volatiles on *M. persicae*, the results obtained herein provide new knowledge. Essential oils and their volatile compounds interact with insect olfactory receptors and therefore could act as repellents [17].

The essential oil of fennel and *Mentha suaveolens* Ehrhart inhibited the settlement of adult apterous individuals of *A. gossypii* in a choice bioassay with zucchini (*Cucurbita pepo* L.) leaves [44]. In addition, it has been found that peppermint (*Mentha x piperita* L.) essential oil and its main compounds menthol and menthone are repellents against apterous individuals of pea aphid *Acyrthosiphon pisum* (Harris), being the most promising substances together with *(E)*-cinnamaldehyde, estragole, carvone and *(E)*-anethole [45]. In our study, *(E)*-anethole (a major compound in anise and fennel essential oil) also showed repellent properties against the winged forms of *M. persicae*. This is consistent with findings [31,46] showing that anise essential oils and their major compound *(E)*-anethole, among others, inhibited the settlement of wingless individuals of *M. persicae* and *M. euphorbiae* in a choice test with pepper leaves. Although winged aphids have more developed olfactory receptors on their antennae than wingless individuals [16], *(E)*-anethole is perceived as a negative signal by both forms (wingless and winged), and hence, it can be considered a promising substance as an aphid repellent.

In field conditions, the potential repellent effect of bioactive volatiles has been tested, and findings showed that the release of garlic extract and *(E)*-β-farnesene was effective for *Metopolophium dirhodum* Walker and *Sitobion avenae* Fabricius, the predominant aphid species in the wheat fields of Belgium [47]. Based on our results from the choice bioassay with potted pepper plants, farnesol (similar to the aphid alarm pheromone, *(E)*-β-farnesene) notably repels the winged forms of *M. persicae*, which confirms previous findings of repellent effects of this substance on winged [30,31] and apterous aphids [31,41,42,46].

On the other hand, spraying methyl esters of coconut fatty acids provided good results both in terms of repellency index and choice distribution. This is the first report on the repellent activity of this product against an aphid crop pest. Some preliminary data were obtained by our group [48] concerning the impact on winged individuals of *M. persicae* of coconut FAME, alone or combined with *(E)*-anethole alone, at a dose of 0.1–0.2% of the active ingredient (nanoemulsion 1:2) sprayed on pepper plants. Previously, the repellent properties of coconut oil-derived compounds were tested [33] against blood-sucking arthropods, such as flies, ticks, bed bugs and mosquitoes in laboratory bioassays. These authors found that their effect can last up to two weeks after application, indicating that these compounds have longer residual activity than that of *N*,*N*-diethyl-meta-toluamide (DEET), considered the standard insect repellent since its development in 1944. It is known that the use of DEET causes health problems in humans, particularly infants and pregnant women, and hence, it is necessary to find alternatives [49]. It has also been shown that an aqueous starch-based formulation containing natural coconut fatty acids protects cattle from fly bites for up to 96 h, which, according to the authors, is the longest reported protection by a natural repellent product [33]. For all these reasons, compounds derived from coconut oil seem to be a good alternative to DEET. It would be interesting to continue investigating the repellent properties of coconut oil-derived compounds on other insect vectors that transmit diseases to humans, animals and plants.

Several veterinary products for flea and tick control in pets contain D-limonene (from citrus peel) as an active ingredient [49]. In aphids, it has been demonstrated by a choice bioassay with pomegranate and grapevine leaf discs that the essential oils of *Citrus x aurantium* L. and *C. reticulata* Blanco showed repellent effects against the pomegranate aphid *Aphis punicae* Passerini and the vine aphid *A. illinoisensis* Shimer [50]. Furthermore, it has been shown that the essential oil of caraway seeds (*Carum carvi* L.), rich in D-carvone and D-limonene, exhibits repellent activity against apterous individuals of *M. persicae* on white cabbage plants (*Brassica oleracea* var. *capitata* L.) [51]. In another study [31], no settlement inhibition of apterous individuals of *M. persicae* or *M. euphorbiae* in pepper leaves associated with lemon oil or its main component, limonene, was observed. Another compound present in citrus essential oils is citral, and it has been demonstrated, through choice bioassays with leaf discs and by recording antennal and body movements, that several organic compounds derived from citral inhibited settlement and had repellent effects against apterous individuals of *M. persicae* [52]. In our experiments, distilled lemon oil was not a good aphid repellent against winged forms of *M. persicae*, but it did improve the repellent properties of the mixtures when added as an ingredient.

The repellent effect of aromatic plant essential oil on aphids was tested [53], and it was reported that the essential oils of *Cymbopogon citratus* (from Candolle) Stapf, *Salvia officinalis* L. and *Origanum majorana* L., repelled apterous individuals of *M. persicae*, *A. gossypii*, *A. spiraecola* Patch and *A. fabae* Scopoli in choice bioassays. On the other hand, it has been observed that the essential oils of two species of *Mentha* sp. and *Salvia* sp. have repellent effects and nymph production deterrence effects against apterous individuals of *A. punicae* in choice bioassays with pomegranate leaves [54].

In the part of the study assessing the reproduction of winged individuals of *M. persicae* on pepper plants after spraying with the bioactive volatile nanoemulsions, promising results were obtained with farnesol + distilled lemon oil + coconut FAME, with this mixture reducing the fecundity of winged individuals of *M. persicae*. Among the possible modes of action of essential oils, the reduction in the fertility of insects [18] should be considered, but in the case of our bioassay, the exposure to the treatment (just 24 h) seems too short a time to be able to appreciate such an effect. Another possible explanation for this reproduction inhibition is that females have doubts about the suitability of the host to establish a colony due to their perception of volatiles. That is, having already landed on the treated plant, they do not like it, perhaps due to its odour or the taste of the plant tissue. Maybe the treatment modifies the composition of the sap and/or cell juices, the insects are confused by the volatiles masking the host plant odour, or they have been made uncomfortable by the residual contact effect of the treatment. Whatever the reasons, it seems that reproduction is inhibited in some instances, but to confirm and understand this effect, there is a need for further research.

## 5. Conclusions

Based on our findings, we can conclude that the methyl esters of coconut fatty acids are slightly repellent to winged individuals of *M. persicae*, which is an innovation in the potential use of this product against an agricultural pest providing the candidate substance is registered as an active ingredient for plant protection.

The pure compounds farnesol and *(E)*-anethole also slightly repelled the winged forms of *M. persicae*. Distilled lemon oil, despite not showing repellent activity on its own, is an interesting substance to consider in a formulation, in blends with other bioactives. More research is needed on the repellent properties of the aforementioned products on aphid pests.

## Figures and Tables

**Figure 1 insects-15-00731-f001:**
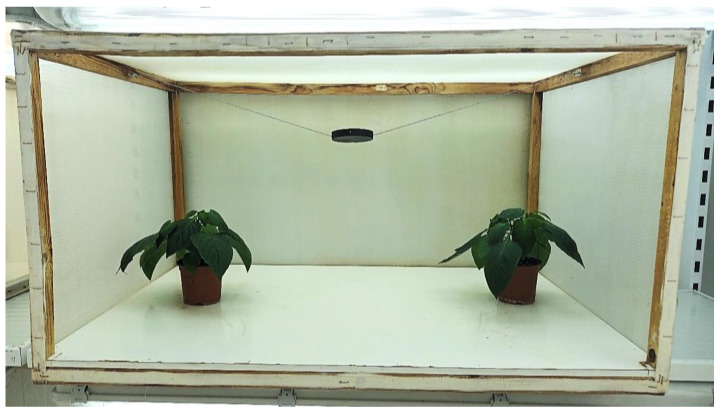
Entomological cage with choice bioassay (control plant and treated plant) and the winged aphid shuttle.

**Figure 2 insects-15-00731-f002:**
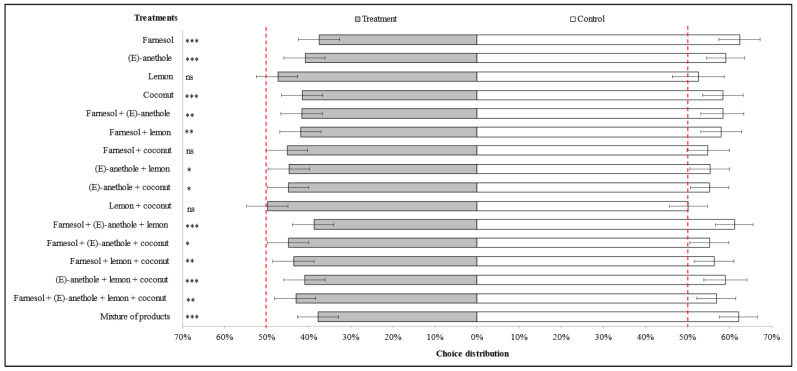
Distribution of choice percentage of aphids (*M. persicae* winged) 24 h after release in cages (*n* = 24) with treated and control pepper plants. The bioactive volatiles were formulated as nanoemulsions at 0.2% or 0.1% in blends of more than one product, with Tween 80. Statistical significance according to an exact two-sided binomial test (* *p* < 0.05; ** *p* < 0.01; *** *p* < 0.001; ns *p* > 0.05). When the confidence interval crosses the red dashed line at 50%, there is no significant repellency or attraction. Error bars correspond to the Clopper–Pearson 95% confidence intervals.

**Table 1 insects-15-00731-t001:** Choice distribution and repellence index (RI) after 24 h of *M. persicae* winged individuals released in cages ^1^ with treated (T) and control (C) pepper plants.

Treatments ^2^	Number of Aphids				% Choice	% No Choice	RI (%) ^3^
	C	T	Choice	No Choice			
Farnesol	245	147	392	208	65.33	34.67	40.24
(*E*)-anethole	282	195	477	123	79.50	20.50	30.85
Lemon	141	127	268	332	44.67	55.33	9.93
Coconut	242	172	414	186	69.00	31.00	28.93
Farnesol + (*E*)-anethole	219	156	375	225	62.50	37.50	28.77
Farnesol + lemon	245	177	422	178	70.33	29.67	27.76
Farnesol + coconut	213	175	388	212	64.67	35.33	17.84
(*E*)-anethole + lemon	249	201	450	150	75.00	25.00	19.28
(*E*)-anethole + coconut	264	214	478	122	79.67	20.33	18.94
Lemon + coconut	255	253	508	92	84.67	15.33	0.78
Farnesol + (*E*)-anethole + lemon	290	184	474	126	79.00	21.00	36.55
Farnesol + (*E*)-anethole + coconut	260	211	471	129	78.50	21.50	18.85
Farnesol + lemon + coconut	247	191	438	162	73.00	27.00	22.67
(*E*)-anethole + lemon + coconut	222	154	376	224	62.67	37.33	30.63
Farnesol + (*E*)-anethole + lemon + coconut	264	200	464	136	77.33	22.67	24.24
Mixture of products ^4^	288	175	463	137	77.17	22.83	39.24

^1^ For each treatment, 24 cages were set up (*n* = 24), and in each cage, 25 winged aphids were released. That is, a total of 600 aphids per treatment. ^2^ O/W nanoemulsions of bioactive volatiles at 0.2% or 0.1% in blends of more than one product, with Tween 80 (1:2 or 1:1:2, 1:1:1:2, 1:1:1:1:2 for the blends). ^3^ Repellency index (RI) = [1 − (T/C)] × 100. Number of *M. persicae* winged on control plants (C) or treated plants (T) 24 h after aphid release. ^4^ The mixture of product is a combination of *(Z)*-jasmone, citral, distilled lemon oil, farnesol and *(E)*-anethole formulated by an undisclosed method in equal amounts, at a dose of 0.2% of the total of the bioactive products.

**Table 2 insects-15-00731-t002:** Reproduction after 24 h of *M. persicae* winged individuals released in cages ^1^ with treated (T) and control (C) pepper plants.

Treatments ^2^	Mean Number of Nymphs/Adult per Day ^3^		*t*-Test	*p*-Value ^4^
	T	C		
Farnesol	0.54 ± 0.10	0.68 ± 0.09	1.841	0.079
*(E)*-anethole	1.16 ± 0.15	0.99 ± 0.13	−1.700	0.103
Lemon	0.40 ± 0.07	0.34 ± 0.09	−0.898	0.379
Coconut	0.55 ± 0.06	0.54 ± 0.08	−0.175	0.863
Farnesol + *(E)*-anethole	0.75 ± 0.12	0.72 ± 0.09	−0.290	0.774
Farnesol + lemon	0.73 ± 0.07	0.79 ± 0.07	0.468	0.645
Farnesol + coconut	0.76 ± 0.15	0.70 ± 0.11	−0.571	0.574
*(E)*-anethole + lemon	0.66 ± 0.07	0.78 ± 0.08	1.235	0.229
*(E)*-anethole + coconut	0.89 ± 0.09	0.78 ± 0.08	−0.941	0.356
Lemon + coconut	0.99 ± 0.12	0.89 ± 0.11	−0.817	0.422
Farnesol + *(E)*-anethole + lemon	0.60 ± 0.09	0.67 ± 0.07	0.901	0.377
Farnesol + *(E)*-anethole + coconut	0.58 ± 0.07	0.57 ± 0.06	−0.266	0.792
Farnesol + lemon + coconut	0.65 ± 0.10	0.97 ± 0.13	2.301	0.031 *
*(E)*-anethole + lemon + coconut	0.44 ± 0.10	0.41 ± 0.05	−0.266	0.793
Farnesol + *(E)*-anethole + lemon + coconut	0.69 ± 0.11	0.66 ± 0.07	−0.424	0.676
Mixture of products ^5^	0.60 ± 0.07	0.52 ± 0.06	−1.095	0.285

^1^ For each treatment, 24 cages were set up (*n* = 24), and in each cage, 25 winged aphids were released. That is, a total of 600 aphids per treatment. ^2^ O/W nanoemulsions of bioactive volatiles at 0.2% or 0.1% in blends of more than one product, with Tween 80 (1:2 or 1:1:2, 1:1:1:2, 1:1:1:1:2 for the blends). ^3^ Mean ± standard error of the mean number of nymphs per adult per day. ^4^ Asterisks indicate significant differences (*p*-value ≤ 0.05) between T (treated plants) and C (control plants). The *t*-test for paired samples was used for comparison of means in T and C within each treatment. ^5^ The mixture of product is a combination of *(Z)*-jasmone, citral, distilled lemon oil, farnesol and *(E)*-anethole formulated by an undisclosed method in equal amounts, at a dose of 0.2% of the total of the bioactive products.

## Data Availability

The original contributions presented in the study are included in the article, further inquiries can be directed to the corresponding author.

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
