# Peer review of "Repellent Effects of Coconut Fatty Acid Methyl Esters and Their Blends with Bioactive Volatiles on Winged Myzus persicae (Sulzer) Aphids (Hemiptera: Aphididae)"

_insects, 2024, doi:10.3390/insects15090731_

Round 1
Reviewer 1 Report (Previous Reviewer 2)
Comments and Suggestions for Authors
Authors improved the manuscript by resubmitting an improved version. I have no further comments.
Reviewer 2 Report (Previous Reviewer 3)
Comments and Suggestions for Authors
The manuscript has been revised as requested and is recommended for acceptance.
Comments on the Quality of English LanguageMinor editing of English language required.
This manuscript is a resubmission of an earlier submission. The following is a list of the peer review reports and author responses from that submission.
Round 1
Reviewer 1 Report
Comments and Suggestions for Authors
- I would like to kindly request the authors and the journal to check for formatting before sending it for peer-review. There are two Tables numbered 1 and two figures numbered Figure 1 making it difficult to understand which result is being discussed. Thus, my comments here are quite limited, and I was not able to go through the whole paper.
- Seems like Figure 1 on Page 8 and Figure 2 represent a subset of data from Table 1 on Page 6. I would request authors to avoid repeating the same data twice. Either statistical significance can be indicated in Table 1 or remove the last three columns from Table 1.
- Figure 2: Please explain how the standard error was calculated in the legend. how many replicates? The standard error should have been also mentioned in Table 1 on Page 6.
Comments on the Quality of English LanguageEnglish is fine and easy to understand.
Author Response
Comment 1: I would like to kindly request the authors and the journal to check for formatting before sending it for peer-review. There are two Tables numbered 1 and two figures numbered Figure 1 making it difficult to understand which result is being discussed. Thus, my comments here are quite limited, and I was not able to go through the whole paper.
Response 1: We agree with this comment. We have corrected the numbering of tables and figures. Moreover, Figure 1 and citations [41] and [42] of the previous version about the repellency index has been removed from the manuscript and submitted as Non-published material.
Comment 2: Seems like Figure 1 on Page 8 and Figure 2 represent a subset of data from Table 1 on Page 6. I would request authors to avoid repeating the same data twice. Either statistical significance can be indicated in Table 1 or remove the last three columns from Table 1.
Response 2: We approve this comment. On the one hand, we eliminated the columns "%C in choice" and "%T in choice" from Table 1, leaving these values represented in Figure 2 together with the Clopper Pearson confidence interval and the level of statistical significance for each treatment. On the other hand, we eliminated Figure 1 of the previous version about repellency index, leaving these values represented in Table 1, but we submitted the figure and legend as Non-published material together with the citations [41] and [42] of the previous version also removed from the manuscript but submitted as Non-published material.
Comment 3: Figure 2: Please explain how the standard error was calculated in the legend. how many replicates? The standard error should have been also mentioned in Table 1 on Page 6.
Response 3: In Figure 2, the standard error has not been calculated. The error bars represent the 95% Clopper Pearson confidence interval obtained from the statistical analysis performed with the R program. In line 222 of section 2.6 statistical analysis of materials and methods, the citation [43] with the R software version used is added. The legend in Figure 2 on line 285 has been modified to clarify that the error bars represent the Clopper Pearson confidence interval. Replicates and repetitions are indicated in the materials and methods section on lines 209-213, 4 replicates or treatment applications per week (Monday-Thursday) and 24 replicates (cages) per treatment. The legend in Figure 2 on line 282 is modified to clarify this.
Reviewer 2 Report
Comments and Suggestions for Authors
Review of manuscript “Repellent effects of coconut fatty acid methyl esters and their blends with bioactive volatiles on winged Myzus persicae (Sulzer) aphids (Hemiptera: Aphididae)”
The manuscript addresses an important topic which is the control of aphids in pepper plants using natural products. However, the scope of the research is limited because the effect was only evaluated after 24 hours and under laboratory conditions. Thus, I suggest to include in the title “.. in laboratory bioassays.”
The compounds evaluated included: (E)-anethole, farnesol, distilled lemon oil (Citromil S.L), coconut FAME (waxy starch-coconut fatty acid methyl ester composite), and a mixture formulated by Idai Nature S.L. Low energy nano emulsions preparations was “Oil in water (O/W) nanoemulsions were prepared using a high-speed dispersion ma-177 chine IKA® Labor-Pilot Controller 2000/4”. Concentrations used were 0.2% alone or 0.1% in blends. Choice experiment between sprayed and non sprayed pepper plants were conducted, and evaluations included: % aphids on plants and calculated RI, offspring at 24h.
A more detailed discussion about the differential results obtained between choice and reproductive output would enrich the manuscript.
Specific comments:
L72-73: Include a reference source for this statement: “Probably due to the evolutionary process, winged individuals have more developed olfactory antennal receptors than wingless individuals.”
L96: word missing: “It has reported” should read “It has been reported”
L104: improper translation for “soya”
L139 replace “a lot” for a more formal expression.
Table 1: “Mean number of nymphs/adult and day”, here might be better to delete “and” or replace by “per”.
Table 1: Does the mixture of products correspond to the mixture formulated by Idai Nature S.L.? please clarify in table footnote to avoid confusion.
Revise figure numbering, because figure 2 (L 280) is presented before figure 1 (L 285).
Figures 1 and 2 contain almost same information presented in Table 1, except for statistics. If statistical info is included in Table 1, these Figures can be deleted. Can you avoid duplicated presentation of results?
L145: write objectives in past tense
**L168-169 “product mixture formulated by Idai Nature S.L. (La Pobla de Vallbona, Valencia, Spain) using an undisclosed method”
L 211 - 212: Cite original source for this formula. To be consistent with citation format, use numbers.
L311: M. persicae should be in italics format
L332: replace “pepper pots” by “potted pepper plants”
L 400: This conclusion: “it enhances the effects by acting as a cosolvent, adjuvant or stabilizer” is not derived from the results presented in this manuscript and should not be included here.
Comments on the Quality of English Language
Past tense format needs review
Author Response
Comment 1: The manuscript addresses an important topic which is the control of aphids in pepper plants using natural products. However, the scope of the research is limited because the effect was only evaluated after 24 hours and under laboratory conditions. Thus, I suggest to include in the title “.. in laboratory bioassays.”
Response 1: We appreciate the suggestion but we consider that the title would be too long. Instead, we have decided to specify in lines 15 and 23 that it has been carried out in a bioassay in culture chambers.
Comment 2: A more detailed discussion about the differential results obtained between choice and reproductive output would enrich the manuscript.
Response 2: We are very grateful for this comment, but we have not found any studies in the literature that relates repellency to aphid reproduction. We hope to expand our knowledge of this in future research.
Comment 3: L72-73: Include a reference source for this statement: “Probably due to the evolutionary process, winged individuals have more developed olfactory antennal receptors than wingless individuals.”
Response 3: The reference source for this statement corresponds to the citation [15]. It is decided to move the citation [15] from line 73 to line 75 to clarify this.
Comment 4: L96: word missing: “It has reported” should read “It has been reported”
Response 4: We agree with this comment. We added "It has been reported" on line 97
Comment 5: L104: improper translation for “soya”
Response 5: We agree with this comment. We correct "soy" on line 105
Comment 6: L139 replace “a lot” for a more formal expression.
Response 6: We agree with this comment. We replaced "a lot" for "much" in line 140
Comment 7: Table 1: “Mean number of nymphs/adult and day”, here might be better to delete “and” or replace by “per”.
Response 7: We agree with this comment. We replace "and" by "per" in Table 2 page 8 in line 300
Comment 8: Table 1: Does the mixture of products correspond to the mixture formulated by Idai Nature S.L.? please clarify in table footnote to avoid confusion.
Response 8: We appreciate this comment. We modified lines 279 and 307 to clarify that the mixture of products treatment has been formulated by an undisclosed method.
Comment 9: Revise figure numbering, because figure 2 (L 280) is presented before figure 1 (L 285).
Response 9: We agree with the comment. We have decided to delete the Figure about the repellency index of the previous version to avoid repeating information with Table 1.
Comment 10: Figures 1 and 2 contain almost same information presented in Table 1, except for statistics. If statistical info is included in Table 1, these Figures can be deleted. Can you avoid duplicated presentation of results?
Comment 10: We approve this comment. On the one hand, we eliminated the columns "%C in choice" and "%T in choice" from Table 1, leaving these values represented in Figure 2 together with the Clopper Pearson confidence interval and the level of statistical significance for each treatment. On the other hand, we eliminated Figure 1 of the previous version about repellency index, leaving these values represented in Table 1, but we submitted the figure and legend as Non-published material together with the citations [41] and [42] of the previous version also removed from the manuscript but submitted as Non-published material.
Comment 11: L145: write objectives in past tense
Comment 11: Agree. We have, accordingly, done the change. Lines 144-149. Rewrite verb tenses in the past
Comment 12: L211 - 212: Cite original source for this formula. To be consistent with citation format, use numbers.
Response 12: Agree. We have, accordingly, done the change. Lines 214. The citation "(Gutierrez et al., 1997)" now corresponds to reference [42]. In the same way, in line 175 with the citation "(Pascual-Villalobos et al. 2017)" it now becomes a reference [41]
Comment 13: L311: M. persicae should be in italics format
Response 13: Agree. We have, accordingly, done the change. Lines 227, 276 and 311, "M. persicae" has been written in italics
Comment 14: L332: replace “pepper pots” by “potted pepper plants”
Response 14: Agree. We have, accordingly, done the change. L332 "pepper pots" has been replaced by “potted pepper plants”
Comment 15: L 400: This conclusion: “it enhances the effects by acting as a cosolvent, adjuvant or stabilizer” is not derived from the results presented in this manuscript and should not be included here.
Response 15: We agree with this comment. We decided to remove that part of the conclusions
Reviewer 3 Report
Comments and Suggestions for Authors
The current paper uses different botanical products as volatile metabolites to induce repellency of aphids, which damage crops and are potential carriers of viral pathogens. The study is novel, and the contents are well-written. I have a concern and a few suggestions regarding the design of the study.
In the author-quoted literature, these botanical products have been reported against different species of aphids, but no study has reported the effect of these oils on the host plant. Since they have an allelochemical effect, besides inhibiting microbial or pathogenic growth or repelling insects, they also affect plant growth. Did the author check their effect on plants?
The summary should stand alone, and here, it seems more descriptive. Please revise it from an analytical perspective, focusing on the experimental approach and the results.
There isn’t any background detail about using nymphs. Please mention it together with insects and plants under methods.
What are the long-term effects of using botanical products for aphid repellency?
Since the author mentioned that the aphids and plants were renewed every day, I'm curious about the intensity of the odour. Did the author systematically measure it to determine if the Repellency Index changed from the first day onward?
Some typos and format mistakes, please go through the whole MS.
lines 227, 275, and 311 M. persicae, should have botanical names italicized.
Comments on the Quality of English LanguageMinor editing of English language required.
Author Response
Comment 1: In the author-quoted literature, these botanical products have been reported against different species of aphids, but no study has reported the effect of these oils on the host plant. Since they have an allelochemical effect, besides inhibiting microbial or pathogenic growth or repelling insects, they also affect plant growth. Did the author check their effect on plants?
Response 1: We appreciate this comment, but as it is not a priority objective of this study, we have not determined how volatile bioactives affect plant growth. Although it is true, no treatment caused phytotoxicity to plants in the doses tested. Future work will try to study how volatile bioactives affect plant growth.
Comment 2: The summary should stand alone, and here, it seems more descriptive. Please revise it from an analytical perspective, focusing on the experimental approach and the results.
Response 2: We appreciate the comment but we consider that the abstract is structured in a balanced way between introduction, methodology, results and conclusions. We prefer not to make any changes to the abstract so as not to exceed the word limit established by the journal.
Comment 3: There isn’t any background detail about using nymphs. Please mention it together with insects and plants under methods.
Response 3: We appreciate this comment. We clarify the information about nymphs in lines 162-163 section 2.1 insects and plants.
Comment 4: What are the long-term effects of using botanical products for aphid repellency?
Response 4: It is a very interesting question but we do not have an answer since in this work the long-term effects of the use of botanical products for aphid repellency have not been studied. It would be interesting to know the long-term effect of the use of botanical products on both aphids and plants, as well as their long-term effect on the environment and on mammals.
Comment 5: Since the author mentioned that the aphids and plants were renewed every day, I'm curious about the intensity of the odour. Did the author systematically measure it to determine if the Repellency Index changed from the first day onward?
Response 5: We do not measure the intensity of the odour at any time. The plants and aphids were renewed daily because we were interested in obtaining a rapid response to the effect (24 h) and to avoid errors in the counts. In the same week, only the same treatment was tested so as not to interfere with the odors of the different bioactive volatiles tested. At the end of the last count, the cages were cleaned and the chamber was ventilated, leaving a period of 3 days between treatments, enough time to renew the air inside the chamber and not to overlap the odors of the different bioactive volatiles tested.
Comment 6: lines 227, 275, and 311 M. persicae, should have botanical names italicized.
Respose 6: Agree. We have, accordingly, done the change. Lines 227, 276 and 311, "M. persicae" has been written in italics.